# Association of serum Klotho with tinnitus prevalence, duration and severity: A cross-sectional study in middle-aged and older adults

Ke-Jiang Du[1,2], Bin-Yu Mo[2], Tao Hou[1], Long Chen[1], Deng-Rong Su[1], Shi-Hua Yin[1]*

1 Department of Otorhinolaryngology Head and Neck Surgery, Second Affiliated Hospital of Guangxi Medical University, Nanning, Guangxi, China, 2 Department of Otorhinolaryngology Head and Neck Surgery, Liuzhou People's Hospital affiliated to Guangxi Medical University, Liuzhou, Guangxi, China

* shihuayin@gxmu.edu.cn

## Abstract

### Background

As an anti-aging protein, although studies increasingly suggest that the Klotho plays a role in the auditory system, the link between serum Klotho levels and tinnitus remains poorly characterized. The aim of this study was to investigate the associations between serum Klotho levels and tinnitus focusing on prevalence, duration and severity in middle-aged and older adults.

### Methods

We performed a population-based cross-sectional study of individuals in the National Health and Nutrition Examination Survey (NHANES) 2009–2012 and 2015–2016. Univariable and multivariable logistic regression was used to evaluate the relationship between serum Klotho levels and tinnitus, with adjustment for potential confounders and further age-stratified analyses. Restricted cubic splines were applied to assess potential non-linearity in the dose-response relationship. Furthermore, Subgroup and interaction analyses were conducted to assess the consistency of this association.

### Results

In this research of 3280 individuals aged 40–79 years with a median age of 55 (IQR: 48, 62) and 48% male participants, the median serum Klotho level was 779.80pg/mL. Multivariable logistic regression uncovered consistent inverse associations between serum Klotho levels and tinnitus incidence across progressively adjusted models (ORs: 0.68–0.70, $p = 0.009$–0.01). Age-based stratified analyses suggested the strongest effect was observed in individuals aged 50–59 years (full model ORs: 0.55, $p = 0.046$). A marginally significant protective effect was observed in the 60–69

**Data availability statement:** The data used for this study can be accessed in the Supporting Information files and through https://ww.cdc.gov/nchs/nhanes.

**Funding:** This work was supported by the National Natural Science Foundation of China (Grant No. 82160213).

**Competing interests:** The authors have declared that no competing interests exist.

age group (unadjusted OR=0.62, 95% CI: 0.39–1.00, $p=0.050$). Serum Klotho levels showed no significant correlation with tinnitus severity, while a potential association with the duration of tinnitus was observed ($p=0.058$). Additionally, restricted cubic spline analysis revealed a linear inverse association between Klotho and tinnitus risk stratified by age (all $p$ for nonlinear >0.15). Finally, subgroup and interaction analyses revealed no significant effect modification (all $p$-interaction >0.1).

## Conclusion

Serum Klotho concentrations showed a consistent inverse association with tinnitus prevalence in US adults, with the strongest effect observed in individuals aged 50–69 years.

---

## 1. Introduction

Tinnitus, a prevalent auditory disorder, is characterized by the persistent experience of phantom sounds (e.g., buzzing, ringing) without any corresponding external acoustic stimulation. Chronic tinnitus, in particular, generally significantly diminishes the quality of life, impairing social function and occupational performance due to its persistent and disturbing noises [1]. Furthermore, it is often associated with psychological issues such as anxiety and depression, leading to significant socioeconomic burdens on healthcare systems. Epidemiological studies demonstrate that tinnitus affects 10–15% of adults globally [2], particularly pronounced among the aging population reaching 30.3% [3], significantly affecting their quality of life. The global burden of tinnitus is expected to rise significantly with population aging. Notably, there is still an absence of objective indicators for the diagnosis of tinnitus. Meanwhile, Current treatments, such as medication, hearing aids, and sound therapy, show limited effectiveness in clinical settings. Therefore, there remains an urgent need for research into novel biomarkers and therapeutic strategies to address this condition effectively.

Age-related factors has been regarded as a major risk factor for tinnitus [4]. While Klotho protein serves as an anti-aging transmembrane protein that functions as both a coreceptor for fibroblast growth factor 23 (FGF23) and a master modulator of calcium-phosphate homeostasis, with demonstrated pleiotropic effects on organismal longevity and multi-organ functional integrity [5–7]. The Klotho family comprises three members (α-, β-, and γ-Klotho) with distinct tissue-specific expression patterns and physiological functions [8]. Among these, α-Klotho has been the most extensively investigated and remains the predominant focus of current research [9]. This study focused specifically on α-Klotho (hereafter termed "Klotho"). Existing literature have suggested that the Klotho protein, which is known to be linked to aging, cognitive function, and various neurological disorders, may play a role in the pathophysiology of auditory system [10–12]. Recently, epidemiological evidence links decreased serum Klotho levels with higher hearing loss risk [13,14]. However, the relationship between serum Klotho levels and tinnitus risk remains poorly characterized.

Therefore, exploring the correlation between serum Klotho and tinnitus phenotypes presents a promising avenue for advancing our understanding of tinnitus' etiology and potential interventions.

In this study, we employed a cross-sectional design to investigate the association between serum Klotho protein levels and tinnitus in a cohort of middle-aged and elderly individuals from National Health and Nutrition Examination Survey (NHANES). By integrating serum biomarker assessments with auditory evaluations, we aimed to elucidate whether serum Klotho correlate with the prevalence, duration and severity of tinnitus among middle-aged and older adults, potentially offering new insights into the biological underpinnings of this auditory disorder. In conclusion, our research seeks to contribute valuable knowledge to the field of otology and geriatrics, and the epidemiological evidence on the role of Klotho in tinnitus.

## 2. Materials and methods

### 2.1 Study population

The National Health and Nutrition Examination Survey (NHANES), conducted by the National Center for Health Statistics, was a nationally representative study to assess the health status of the noninstitutionalized U.S. civilian population. NHANES employed a multistage probability oversampling design to collect nationally representative data via interviews, physical examinations, and laboratory tests [15]. Detailed survey methodology, design, and results were publicly available. The NHANES study protocol was reviewed and approved by the National Center for Health Statistics (NCHS) Institutional Review Board (IRB), and written informed consent was obtained from all participants prior to data collection [16]. This study utilized de-identified, publicly available data from the NHANES database, and the authors had no access to identifiable participant information. Therefore, our study was exempt from additional ethical review. The data were accessed on between February 15 and April 10, 2025. Although there were serum Klotho data for participants from 40 to79 years in 2007–2016, we did not cover the years 2007–2008 and 2013–2014 as the relevant audiometry data was not available. Therefore, our research utilized NHANES data from the residual three cycles. The analytical sample size was determined by the available NHANES dataset (https://wwwn.cdc.gov/nchs/nhanes/). This study complied with the STROBE (Strengthening the Reporting of Observational Studies in Epidemiology) statement guidelines.

### 2.2 Exposure and outcome variables

Serum Klotho protein was the primary exposure variable of interest. all measurements were performed according to the laboratory's standardized criteria before being issued. Specimens were stored at – 80°C at the Centers for Disease Control and Prevention in Atlanta, GA, where they were shipped on dry ice to the Northwest Lipid Metabolism and Diabetes Research Laboratories at the University of Washington in Seattle, WA. The serum Klotho levels were assessed using ELISA (IBL-International, Japan), following the manufacturer's protocol. Further details on laboratory methodology, quality assurance, and monitoring can be found at the following link:

https://www.cdc.gov/Nchs/Data/Nhanes/Public/2009/DataFiles/SSKL_F.htm.

The main outcome variable was tinnitus, assessed by asking participants, which was defined as "bothersome tinnitus in the past 12 months," replying "yes" to the question: "In the past 12 months, have you experienced ringing, roaring, or buzzing in your ears or head that lasted for 5 minutes or more? [4,17]" Individuals with tinnitus were asked two follow-up questions: (a) Severity: How much of a problem is the ringing (no problem, small, moderate, big, or very big problem)? (b) Duration: How long have you been bothered by the ringing or roaring (<3 months, 3 months-1 year, 1–4 years, 5–9 years, or ≥10 years)? Based on their responses, Tinnitus duration was classified as "non-chronic" if it lasted less than one year and "chronic" if it lasted one year or longer; Tinnitus severity was categorized into three levels: "none", "mild" and "moderate to severe".

## 2.3  Other covariates

In accordance with current research and biological plausibility, we examined a range of potential covariates, including age, gender, ethnic origin, education level, marital status, body mass index (BMI), poverty income ratio (PIR), smoking status, alcohol consumption status, total cholesterol, noise exposure, Patient Health Questionnaire-9 for depression (PHQ-9), pure-tone average (PTA) and comorbid conditions such as diabetes, hypertension, and cardiovascular disease, as identified in the literature.

In this research, age was considered as a continuous variable. Participants' self-reported ethnic origin involved four categories: non-Hispanic white, non-Hispanic black, Mexican American and other Hispanics. Education levels were categorized into three groups: below high school, completed high school and beyond high school. Marital status was categorized into "single, divorced or widowed" and "married or living with partner". BMI (kg/m$^2$) was categorized as "<25.0", "25-29.9" and "≥30.0". Family income to poverty ratios (PIR) were classified as "≤1" and ">1". Smoking was defined as smoke at least 100 cigarettes in one's life. Alcohol consumption status was determined by whether they had consumed at least 12 alcohol drinks in a year, those responding affirmatively were deemed as alcohol drinkers. A total cholesterol of at least 240 mg/dL is defined as hypercholesterolemia. Noise exposure was defined as a positive endorsement of either item assessing occupational or recreational noise exposure in audiometric questionnaire. Pure-tone average (PTA) was utilized to define hearing loss, which was defined as PTA thresholds ≥25 dB HL at 500, 1, 2, and 4 kHz, as previously mentioned [18,19]. Depression severity was assessed using the PHQ-9 total score, calculated by summing responses across all nine items. Using a validated cutoff of 10, PHQ-9 scores were categorized as "not depressed" (scores <10) or "depressed" (scores ≥10). Diabetes and hypertension, as well as cardiovascular disease including congestive heart failure, coronary heart disease, angina pectoris, heart attack and stroke, were defined as a self-reported physician diagnosis of these conditions. Responses such as "Refused" or "Don't know" to any question were regarded as missing data.

## 2.4  Statistical analysis

Sample weights were computed and applied in all analyses following NCHS methodological recommendations using R software version 4.4.3 (http://www.R-project.org). Variables were classified as continuous or categorical. For descriptive statistics, continuous variables were expressed as weighted medians with interquartile ranges (Q1, Q3), while categorical variables were reported as weighted numbers (percentages). Standardized mean differences (SMD) were calculated by dividing the weighted mean difference by the pooled weighted standard deviation. Survey weights were incorporated in all SMD calculations to account for the complex sampling design. Weighted linear regression and weighted chi-square tests were used to calculate P-values, respectively. In this study, both serum Klotho levels and age showed non-normal distributions, as indicated by the Shapiro-Wilk test ($p < 0.001$). This finding was further confirmed by density plots and Q-Q plots (S1 Fig).

Logistic regression was leveraged to estimate odds ratios (ORs) with 95% confidence intervals (CIs) for the association between serum Klotho and tinnitus by multiple models. Owing to the non-normal distribution, participants were stratified into three groups based on the lower quartile and median of serum Klotho concentrations (Low: ≤ 648.75pg/mL; Medium: 648.76 ~ 779.80pg/mL; High: > 779.80pg/mL). Moreover, A natural logarithm(ln) transformation was applied to serum Klotho levels to normalize their distribution. Meantime, nonlinear relationships of serum Klotho with tinnitus incidence were modeled using restricted cubic splines (RCS). To address age as a potential confounder, we assessed the Klotho-tinnitus relationship through age-stratified logistic regression analyses. Subgroup analyses were further performed to assess heterogeneous associations between serum Klotho levels and tinnitus across populations, and subgroup interactions were assessed via likelihood ratio tests. Statistical significance was defined as $p < 0.05$.

## 3. Results

### 3.1 Characteristics of the participants

The complete-case analysis, following exclusion of missing values for serum Klotho, tinnitus, and covariates, included 3,280 eligible participants (**Fig 1**). In this study, serum Klotho concentrations ranged from 152.50 to 5038.30pg/mL, with a median of 779.80pg/mL (IQR: 648.75–1001.25pg/mL). **Table 1** presents the demographic and clinical characteristics of participants stratified by tinnitus status. Significant differences were observed in all characteristics by tinnitus status, except for BMI, educational level, poverty-income ratio, alcohol consumption status, smoking status, diabetes and total cholesterol levels. Compared to non-tinnitus participants, those with tinnitus were more likely to be older, male, of non-Hispanic white ethnicity, and in a single/divorced/widowed marital status, as well as having hypertension, cardiovascular disease, noise exposure, depression, and hearing loss. Similarly, inverse associations between serum Klotho concentrations and tinnitus prevalence were observed consistently across both continuous and categorical variable analyses.

Median [IQR] for nonnormally distributed continuous variables: P value was calculated by weighted linear regression model. Number (%) for categorical variables: The P value was calculated by weighted chi-square test. Standardized mean differences (SMD) were calculated by dividing the weighted mean difference by the pooled weighted standard deviation. The classification of serum Klotho was based on the unweighted lower quartile and median of concentrations (Low: ≤648.75pg/mL; Medium: 648.76~779.8pg/mL; High: >779.8pg/mL). All estimates were weighted to be nationally representative. Abbreviations: BMI: body mass index; PIR: poverty income ratio; T-cholesterol: Total cholesterol; PHQ-9: Patient Health Questionnaire-9; PTA: Pure-tone average.

A total of 691 tinnitus patients with complete data were included in the further analysis of tinnitus duration and severity. As shown in **Table 2**, A potential association was found between Klotho and tinnitus duration ($p = 0.058$). However, no significant association was observed between Klotho levels and tinnitus severity (S1 table).

### 3.2 Univariate analysis of tinnitus

As shown in S2 Table, the factors of age, BMI, marital status (single, divorced or widowed), smoking, diabetes, hypertension, cardiovascular disease, noise exposure, depression, and hearing loss all show significant positive correlations with

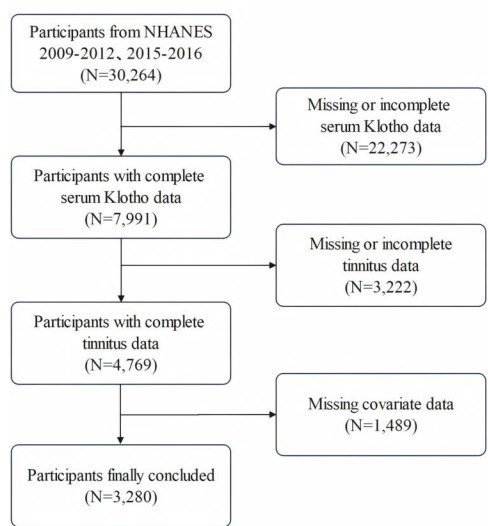

**Fig 1. Participant screening flowchart.**

**Table 1. Baseline characteristics of participants.**

| Characteristic | Overall (n = 3,280) | Tinnitus | | SMD | *P*-value |
|---|---|---|---|---|---|
| | | No (n = 2,587) | Yes (n = 693) | | |
| **Age** (years) (median [IQR]) | 55 [48, 62] | 54 [47, 62] | 57 [50, 63] | −0.225 | <0.001 |
| **Gender**, n (%) | | | | 0.165 | 0.009 |
| Female | 1,701 (52) | 1,360 (54) | 341 (46) | | |
| Male | 1,579 (48) | 1,227 (46) | 352 (54) | | |
| **Ethnic origin**, n (%) | | | | 0.094 | 0.018 |
| Non-Hispanic White | 1,492 (80) | 1,138 (79) | 354 (83) | | |
| Non-Hispanic Black | 831 (9.1) | 690 (9.7) | 141 (7.0) | | |
| Mexican American | 527 (6.4) | 407 (6.5) | 120 (6.2) | | |
| Other Hispanic | 430 (4.9) | 352 (5.2) | 78 (3.7) | | |
| **BMI** (kg/m²), n (%) | | | | 0.146 | 0.094 |
| <25 | 642 (22) | 518 (22) | 124 (23) | | |
| 25-29.9 | 1,105 (34) | 898 (36) | 207 (29) | | |
| ≥30 | 1,533 (43) | 1,171 (42) | 362 (48) | | |
| **Educational level**, n (%) | | | | 0.132 | 0.13 |
| Below high school | 784 (13) | 602 (12) | 182 (15) | | |
| Completed high school | 779 (21) | 608 (20) | 171 (25) | | |
| Beyond high school | 1,717 (66) | 1,377 (67) | 340 (61) | | |
| **Marital status**, n (%) | | | | 0.143 | 0.009 |
| Single, divorced or widowed | 1,215 (30) | 929 (28) | 286 (35) | | |
| Married or living with partner | 2,065 (70) | 1,658 (72) | 407 (65) | | |
| **PIR**, n (%) | | | | 0.027 | 0.5 |
| ≤1 | 647 (11) | 485 (10) | 162 (11) | | |
| >1 | 2,633 (89) | 2,102 (90) | 531 (89) | | |
| **Alcohol status**, n (%) | | | | 0.039 | 0.5 |
| No | 866 (19) | 683 (18) | 183 (20) | | |
| Yes | 2,414 (81) | 1,904 (82) | 510 (80) | | |
| **Smoking status**, n (%) | | | | −0.099 | 0.12 |
| No | 1,658 (51) | 1,357 (52) | 301 (47) | | |
| Yes | 1,622 (49) | 1,230 (48) | 392 (53) | | |
| **Diabetes**, n (%) | | | | 0.112 | 0.084 |
| No | 2,572 (84) | 2,058 (85) | 514 (81) | | |
| Borderline | 101 (2.8) | 77 (2.6) | 24 (3.4) | | |
| Yes | 607 (13) | 452 (12) | 155 (16) | | |
| **Hypertension**, n (%) | | | | −0.145 | 0.045 |
| No | 1,697 (60) | 1,404 (61) | 293 (54) | | |
| Yes | 1,583 (40) | 1,183 (39) | 400 (46) | | |
| **T-cholesterol** (mg/dL), n (%) | | | | 0.014 | 0.8 |
| <240 | 2,756 (82) | 2,170 (82) | 586 (82) | | |
| ≥240 | 524 (18) | 417 (18) | 107 (18) | | |
| **Cardiovascular disease**, n (%) | | | | −0.160 | <0.001 |
| No | 2,872 (92) | 2,311 (93) | 561 (87) | | |
| Yes | 408 (8.5) | 276 (7.3) | 132 (13) | | |
| **PHQ-9**, n (%) | | | | 0.268 | <0.001 |
| <10 | 2,948 (92) | 2,382 (94) | 566 (85) | | |
| ≥10 | 332 (7.7) | 205 (5.8) | 127 (15) | | |

*(Continued)*

**Table 1.** (Continued)

| Characteristic | Overall (n = 3,280) | Tinnitus | | SMD | P-value |
|---|---|---|---|---|---|
| | | No (n = 2,587) | Yes (n = 693) | | |
| **Noise exposure**, n (%) | | | | | <0.001 |
| No | 1,956 (63) | 1,627 (67) | 329 (50) | −0.340 | |
| Yes | 1,324 (37) | 960 (33) | 364 (50) | | |
| **PTA**, dB, n (%) | | | | 0.516 | <0.001 |
| <25 | 2,252 (72) | 1,902 (77) | 350 (53) | | |
| ≥25 | 1,028 (28) | 685 (23) | 343 (47) | | |
| **Serum klotho** (pg/mL), (median [IQR]) | 793 [658, 989] | 805 [664, 992] | 765 [638, 951] | 0.090 | 0.035 |
| **Serum klotho**, n (%) | | | | 0.184 | 0.011 |
| Low | 820 (24%) | 616 (23%) | 204 (28%) | | |
| Medium | 820 (27%) | 634 (26%) | 186 (31%) | | |
| High | 1,640 (49%) | 1,337 (51%) | 303 (42%) | | |

**Table 2. Associations between serum Klotho and tinnitus duration.**

| Characteristic | Tinnitus duration | | | p-value |
|---|---|---|---|---|
| | Overall (n = 691) | Non-chronic (n = 187) | Chronic (n = 504) | |
| **Klotho** (pg/mL) (median [IQR]) | 765 [638, 951] | 823 [685, 1,013] | 760 [629, 930] | 0.058 |
| **Serum Klotho**, n (%) | | | | 0.090 |
| Low | 203 (28%) | 45 (20%) | 158 (29%) | |
| Medium | 186 (31%) | 52 (28%) | 134 (31%) | |
| High | 302 (42%) | 90 (52%) | 212 (39%) | |

Median [IQR] for nonnormally distributed continuous variables. Tinnitus duration was categorized as non-chronic (<1 year) or chronic (≥1 years). The classification of serum Klotho was based on the unweighted lower quartile and median of concentrations (Low: ≤648.75pg/mL; Medium: 648.76~779.8pg/mL; High: >779.8pg/mL). All estimates were weighted to be nationally representative.

tinnitus. Conversely, the non-Hispanic black, the other Hispanic, PIR and serum Klotho were significantly and negatively correlated with tinnitus. In addition, gender, the Mexican American, educational level, alcohol consumption status and total cholesterol showed no significant association with tinnitus.

### 3.3 Associations between serum Klotho and tinnitus

Three progressively adjusted logistic regression models evaluated the serum Klotho-tinnitus association, with covariate adjustment stages detailed in **Table 3**.

Compared to the low-concentration reference group, participants with high-concentration exhibited a significant negative correlation with tinnitus incidence in the unadjusted model (OR =0.68, 95% CI: 0.51–0.90, $P=0.009$). The inverse association persisted after adjustment for demographic and socioeconomic covariates (OR =0.70, 95% CI: 0.52–0.94, $P=0.019$). Additionally, after further adjusting for BMI, alcohol use, smoking, total cholesterol, diabetes, hypertension, cardiovascular disease, noise exposure, hearing loss and depression, the association still maintained significant (OR =0.67, 95% CI: 0.50–0.92, $P=0.01$). Although continuous variable ln (Klotho) showed no statistically significant association with tinnitus ($P=0.08$), the effect direction was consistent with categorical analyses, suggesting a potential dose-response pattern.

 

**Table 3. Multivariable logistic regression model of serum Klotho and tinnitus.**

| Characteristic | ln (Klotho) (n = 3280) | | Serum Klotho Concentration (pg/mL) | | | | | |
|---|---|---|---|---|---|---|---|---|
| | | | Low (n = 820) | Medium (n = 820) | | High (n = 1640) | | |
| | OR (95%CI) | *P*-value | OR (95%CI) | OR (95%CI) | *P*-value | OR (95%CI) | *P*-value | |
| **Unadjusted** | 0.72 (0.50,1.04) | 0.080 | Ref | 0.96 (0.69,1.32) | 0.785 | 0.68 (0.51,0.90) | **0.009** | |
| **Model 1** | 0.78 (0.53,1.14) | 0.186 | Ref | 0.97 (0.70,1.35) | 0.849 | 0.70 (0.52,0.94) | **0.019** | |
| **Model 2** | 0.75 (0.49,1.15) | 0.180 | Ref | 0.90 (0.64,1.27) | 0.540 | 0.67 (0.50,0.92) | **0.010** | |

All estimates were weighted to be nationally representative. ln (Klotho): natural log-transformed serum Klotho concentrations. The classification of serum Klotho was based on the unweighted lower quartile and median of concentrations (Low: ≤ 648.75pg/mL; Medium: 648.76~779.8pg/mL; High: > 779.8pg/mL). Model 1: Adjusted for sociodemographic variables (age, gender, ethnic origin, educational level, marital status and PIR). Model 2: Adjusted for Model 1 + BMI, alcohol status, smoking status, total cholesterol, diabetes, hypertension, cardiovascular disease, noise exposure, hearing loss and depression. Abbreviations: OR, odds ratio; 95% CI, 95% confidence interval.

### 3.4 Associations between serum Klotho and tinnitus stratified by age

Stratified analyses revealed significant age-dependent associations between serum Klotho levels and tinnitus prevalence (Table 4). In adults aged 50−59 years, high Klotho levels (vs. low) were associated with a 45% reduction in tinnitus risk in the unadjusted model (OR=0.55, 95% CI: 0.31–0.97, *p* = 0.038), with a similar trend in the fully adjusted model (OR=0.55, 95% CI: 0.31–0.99, *p* = 0.046). A marginally significant protective effect was observed in the 60−69 age group (unadjusted OR=0.62, 95% CI: 0.39–1.00, *p* = 0.050), though this association attenuated after multivariable adjustment (OR=0.66, 95% CI: 0.37–1.19, *p* = 0.147). Notably, p for trend was significant in both the 50−59 (*p* = 0.031 unadjusted; *p* = 0.048

**Table 4. Association between serum Klotho and tinnitus stratified by age.**

| Age Group | Serum Klotho Concentration (pg/mL) | | | | | | P for trend |
|---|---|---|---|---|---|---|---|
| | Low | Medium | | High | | | |
| | OR (95% CI) | OR (95% CI) | *P*-value | OR (95% CI) | *P*-value | | |
| Unadjusted model | | | | | | | |
| 40-49 | Ref | 1.44 (0.77, 2.70) | 0.246 | 0.79 (0.40, 1.56) | 0.485 | | 0.265 |
| 50-59 | Ref | 0.82 (0.45, 1.48) | 0.495 | 0.55 (0.31, 0.97) | **0.038** | | **0.031** |
| 60-69 | Ref | 0.94 (0.51, 1.75) | 0.846 | 0.62 (0.39, 1.00) | **0.050** | | **0.037** |
| 70-79 | Ref | 0.92 (0.48, 1.75) | 0.795 | 0.97 (0.50, 1.87) | 0.915 | | 0.938 |
| Full model | | | | | | | |
| 40-49 | Ref | 1.43 (0.69, 2.95) | 0.297 | 0.78 (0.33, 1.86) | 0.541 | | 0.353 |
| 50-59 | Ref | 0.62 (0.30, 1.27) | 0.169 | 0.55 (0.31, 0.99) | **0.046** | | **0.048** |
| 60-69 | Ref | 1.03 (0.51, 2.09) | 0.924 | 0.66 (0.37, 1.19) | 0.147 | | 0.113 |
| 70-79 | Ref | 0.73 (0.37, 1.42) | 0.334 | 1.06 (0.55, 2.05) | 0.863 | | 0.757 |

All estimates were weighted to be nationally representative. The classification of serum klotho was based on the unweighted lower quartile and median of concentrations (Low: ≤ 648.75pg/mL; Medium: 648.76~779.8 pg/mL; High: > 779.8pg/mL). Full model: Adjusted for all variables (age, gender, Ethnic origin, educational level, marital status, PIR, BMI, alcohol status, smoking status, total cholesterol, diabetes, hypertension, cardiovascular disease, noise exposure, hearing loss and depression). Abbreviations: OR, odds ratio; 95% CI, 95% confidence interval.

adjusted) and 60−69 (*p* = 0.037 unadjusted) groups, suggesting a dose-response relationship. No significant associations were found in younger (40–49) or older (70−79) cohorts (*p* > 0.05 for all comparisons). Furthermore, restricted cubic spline was utilized to further analysis the nonlinear association between Serum Klotho and tinnitus risk (all *p* for nonlinear >0.15). Although a borderline trend (*p* for overall: 0.06–0.105) in 50–69 years groups, the most pronounced risk reduction occurred at higher concentrations (>779.8 pg/mL) in these individuals (Fig 2).

### 3.5 Subgroup analyses

Subgroup and interaction analyses evaluated potential effect modification in the serum Klotho-tinnitus association across various relevant subgroups. **Fig 3** demonstrates consistent associations across subgroups categorized by age, gender, ethnic origin, smoking, marital status, hypertension, cardiovascular disease, noise exposure, hearing loss and depression. Interaction analyses detected no significant effect modification across subgroups (all *P*-interaction >0.10).

## 4. Discussion

This study utilized data from the three cycles of NHANES (2009–2012 and 2015–2016), focusing on participants aged 40–79 years, to examine the correlation between serum Klotho concentrations and tinnitus prevalence as well as the duration and severity. To our knowledge, this represents the first epidemiological investigation of the association between Klotho and tinnitus, specifically within middle-aged and older individuals.

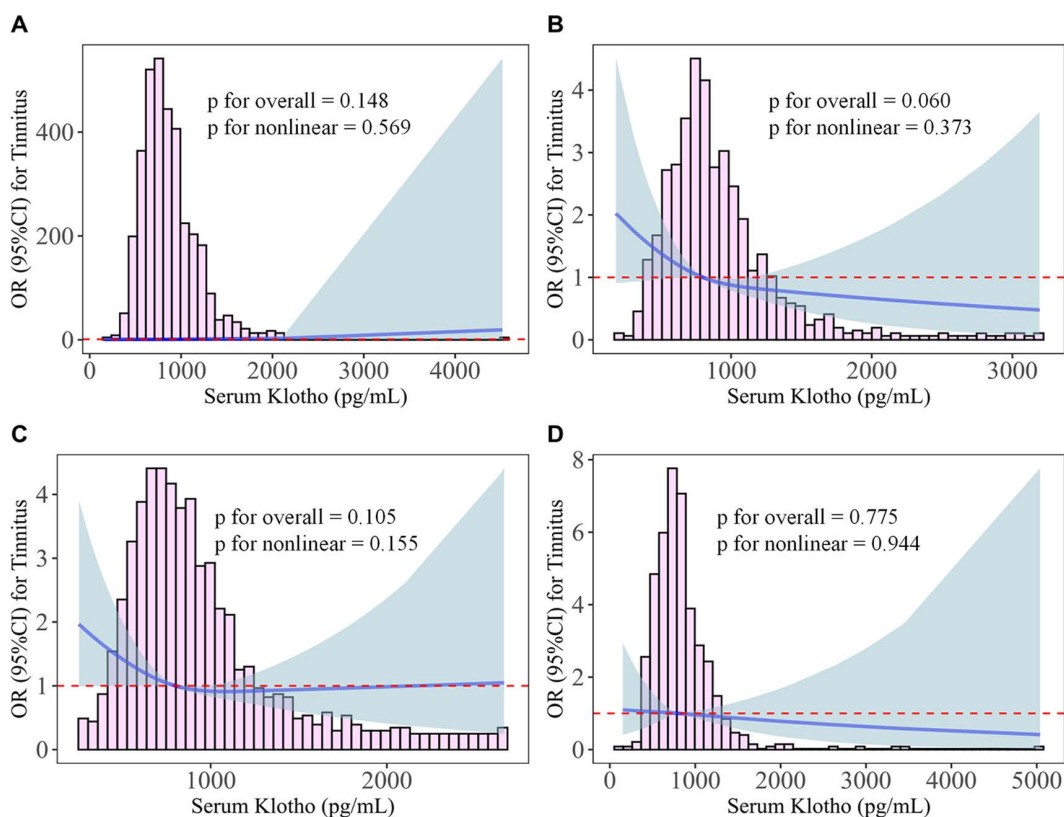

**Fig 2. Dose–response relationship between serum Klotho and tinnitus prevalence stratified by age.** Restricted cubic spline (RCS) with 3 knots was applied to the flexible regression model stratified by age: A. 40–49years, B. 50–59years, C. 60–69years, D. 70–79years. The red dotted line indicates the reference line (OR=1), The blue solid lines and light-blue shaded areas were the ORs for tinnitus and 95% CI, respectively.

| Subgroup | No. of participants | OR(95%CI) | p for Interaction |
|---|---|---|---|
| **Age (years)** | | | 0.376 |
| 40−49 | 812 | 1.72(0.94,3.15) | |
| 50−59 | 850 | 0.79(0.49,1.25) | |
| 60−69 | 965 | 0.76(0.50,1.16) | |
| 70−79 | 653 | 0.83(0.50,1.39) | |
| **Gender** | | | 0.904 |
| Female | 1701 | 0.83(0.58,1.18) | |
| Male | 1579 | 0.9(0.64,1.26) | |
| **Ethnic origin** | | | 0.931 |
| Non−Hispanic White | 1492 | 0.79(0.57,1.11) | |
| Non−Hispanic Black | 831 | 0.83(0.48,1.43) | |
| Mexican American | 527 | 1.14(0.63,2.08) | |
| Other Hispanic | 430 | 1.1(0.5,2.42) | |
| **Smoking** | | | 0.921 |
| No | 1658 | 0.93(0.64,1.35) | |
| Yes | 1622 | 0.86(0.63,1.18) | |
| **Marital status** | | | 0.511 |
| Single,divorced,or widowed | 1215 | 1.01(0.69,1.48) | |
| Married or living with partner | 2065 | 0.81(0.59,1.1) | |
| **Hypertension** | | | 0.701 |
| No | 1697 | 0.88(0.61,1.27) | |
| Yes | 1583 | 0.89(0.64,1.23) | |
| **Cardiovascular disease** | | | 0.596 |
| No | 2872 | 0.85(0.65,1.11) | |
| Yes | 408 | 1.08(0.6,1.94) | |
| **PTA (dB)** | | | 0.854 |
| <25 | 2250 | 0.96(0.69,1.33) | |
| >=25 | 1028 | 0.78(0.54,1.11) | |
| **PHQ−9** | | | 0.602 |
| <10 | 2948 | 0.85(0.66,1.11) | |
| >=10 | 332 | 1.21(0.61,2.38) | |
| **Noise exposure** | | | 0.23 |
| No | 1956 | 0.9(0.64,1.29) | |
| Yes | 1324 | 0.89(0.64,1.25) | |

0.5   1   1.5

**Fig 3. Subgroup analyses of the association between serum Klotho level and tinnitus.** Odds ratios (ORs) were adjusted for age, gender, ethnic origin, educational status, marital status, PIR, BMI, smoking, diabetes, hypertension, cardiovascular disease, total cholesterol, PTA, PHQ-9 and noise exposure. For all subgroup analyses, the stratification variable was excluded from adjusted models. Interaction p-values were derived from likelihood ratio tests (LRT). All estimates were weighted to be nationally representative. Abbreviations: OR: odds ratio; CI: confidence interval; BMI: body mass index; PIR: poverty income ratio; PHQ-9: Patient Health Questionnaire-9; PTA: Pure-tone average.

Our study identified multiple potential risk factors for tinnitus, including age, BMI, single/divorced/widowed marital status, smoking, diabetes, hypertension, cardiovascular disease, noise exposure, depression and hearing loss and lower serum Klotho levels (Table 2), which are consistent with previous studies except for lower serum Klotho [20,21]. In analyzing tinnitus duration, we found that the non-chronic tinnitus group had a higher proportion of elevated Klotho levels, while the chronic tinnitus group had a greater prevalence of reduced Klotho levels, suggesting a potential inverse relationship. Although the correlations did not reach statistical significance ($p = 0.058$), the borderline p-value may indicate clinically meaningful associations. The finding suggests a possible link between serum Klotho levels and tinnitus duration, which warrants further investigation, despite potential limitations due to our sample size. Furthermore, our analysis found no significant association between Klotho levels and tinnitus severity. We hypothesize distinct mechanistic roles of Klotho in the prevalence and duration of tinnitus.

We also observed that participants with high levels of serum Klotho were significantly less likely to report tinnitus than those with low levels of serum Klotho. Our findings suggested that higher serum Klotho concentrations may protect against tinnitus in middle-aged adults (50–69 years), with the strongest effect observed in the 50–59 age group. The age-specific associations align with evidence that Klotho's neuroprotective and anti-inflammatory properties may mitigate age-related auditory decline, particularly during early senescence[22,23]. The loss of significance in the fully adjusted

model for the 60–69 group may reflect confounding by comorbidities (e.g., hypertension, diabetes) or reduced statistical power. The absence of associations in older adults (70–79) could indicated a ceiling effect of age-related cochlear damage, where Klotho's benefits are overridden by irreversible pathology. However, the null results in younger adults (40–49) suggested that Klotho's protective effects may be contingent on age-related physiological changes. Compared to age-stratified models, the apparent attenuation of age subgroup's effect sizes in subgroup analyses likely reflects methodological distinctions: (1) in stratified analyses, covariate effects may be partially offset within age strata, while (2) the interaction model's assumption of uniform covariate effects across age groups could obscure stratum-specific confounding. Importantly, both analytical approaches consistently demonstrated maximal protective effects in middle-aged adults (50–69 years), enhancing the biological plausibility of this age-dependent association. Consequently, these associations remained significant and robust in 50–59 age group after adjusting for confounders and other covariates, suggesting that Klotho may exert independent effects on the prevalence of tinnitus. Our findings uniquely uncovered the significant negative correlation between serum Klotho levels and tinnitus prevalence in a human population, complementing animal studies that suggested a neuroprotective role of Klotho proteins in auditory pathways [10].

Extensive evidence have demonstrated an inverse correlation between serum Klotho concentrations and advancing age [24,25]. Previous studies have indicated that serum Klotho is significantly associated with auditory disorder. Serum Klotho levels were inversely associated with speech- and high-frequency hearing loss risk in adults aged 40–69 years [13]. A similar correlation was discovered in 70–79 years individuals [14]. However, the association between serum Klotho levels and tinnitus remains unexplored. Tinnitus, as a prevalent auditory symptom, is particularly pronounced in the elderly. While prior research reported a tinnitus prevalence of 30.3% in the elderly [26,27], our findings revealed a lower prevalence of 21.1% among middle-aged and older adults, indicating an age-dependent progression of this condition. The global burden of tinnitus is expected to rise significantly with population aging. Current therapeutic interventions demonstrate suboptimal clinical outcomes. Consequently, a pressing need for further research into novel biomarkers and therapeutic approaches to better understand the pathophysiology of tinnitus and improve patient outcomes.

As a crucial component of the FGF23 signaling pathway, Klotho could play a pivotal role in the pathogenesis of age-associated disorders via the FGF-23–Klotho axis, including cardiovascular disease, renal dysfunction, and metabolic syndromes [5,28–30]. Similarly, a recent study suggested that the FGF-23-Klotho endocrine axis may play a novel role in the inner ear regulating directly or indirectly the kinocilium-stereocilia interaction[31], and auditory disorder and middle ear malformations were also showed in FGF23-deficient mice [11]. Animal models confirm that Klotho knockout leads to significant auditory function disorders, and Klotho protein may play an essential role in auditory physiology by preserving endolymphatic ion homeostasis. Klotho protein exhibited predominant expression in both mouse cochlear tissues (stria vascularis and spiral ligament) and renal distal convoluted tubules, potentially suggesting a common physiological role in modulating ion transport mechanisms [12]. Premature age-related hearing loss (ARHL) development in Klotho -/- mice demonstrated a protective role of Klotho protein in preserving auditory function and delaying hearing loss progression [10]. Normalization of serum 1,25-dihydroxyvitamin D3(1,25(OH)2D3) levels via vitamin D-deficient diet can rescues auditory function in Klotho knockout mice with mild hearing loss [32]. Additionally, Klotho downregulation decreases transient receptor potential vanilloid 5(TRPV5) and TRPV6 levels, altering inner ear $Ca^{2+}$ regulation and causing sensory cell transduction defects that manifest as auditory or vestibular disorders [33]. Meanwhile, Klotho reduction may also has affected on activity and protein abundance of KCNQ1/KCNE1, a $K^{(+)}$ channel required for proper hearing, contribute to auditory disorders [34]. These results suggest that Klotho-mediated auditory impairment is linked to dysregulation in the FGF23 signaling pathway, potassium and calcium homeostasis, vitamin D metabolism, and oxidative stress.

The anti-aging properties of Klotho have stimulated extensive investigation into factors modulating its expression and activity. An increasing number of experimental evidences have demonstrated that antioxidant supplementation could

upregulates Klotho protein expression [35–38]. Significantly, population-based evidence also indicated a significant positive correlation between dietary antioxidant (e.g., Vitamin C) intake and serum Klotho levels [39]. Similarly, physical exercise elevates serum Klotho levels in sedentary individuals, potentially mediated by enhanced vagal activity and reduced sympathetic tone [40]. Furthermore, in vivo evidence has supported the therapeutic potential of exogenous recombinant Klotho protein recently [41,42]. Therefore, targeted modulation of Klotho expression represents a promising therapeutic strategy for tinnitus prevention and treatment. Our findings demonstrate the relationship between serum Klotho levels and tinnitus prevalence, suggesting that targeted manipulation of circulating Klotho may represent a viable therapeutic strategy. The potential of Klotho-level modulation may serve as a future research direction in tinnitus.

However, this study has several limitations that must be acknowledged: (1) Tinnitus was defined using standardized questionnaire responses from participant interviews. While this approach aligns with prior NHANES studies, which may introduced potential recall bias. (2) Although our sample size was relatively large, the cross-sectional design restricts the establishment of causal relationships between serum Klotho protein levels and tinnitus. (3) The absence of longitudinal data limits our ability to track changes over time and their potential implications for tinnitus development. (4) Our findings are based primarily on a North American population, which may affect generalizability given potential ethnic variations in Klotho biology. Future multi-ethnic studies (e.g., Asian populations) are needed to validate these.

## 5. Conclusion

Taken together, high serum Klotho levels are associated with reduced tinnitus prevalence in adults aged 50–59 years, with suggestive evidence in those aged 60–69. Significantly, a potential inverse relationship was also uncovered between Klotho and the duration of tinnitus. These findings enhance our pathophysiological understanding of Klotho in tinnitus and suggest the possibility and necessity for intervention studies targeting Klotho pathways. Prospective studies and controlled clinical trials are still required to further elucidate the causal interpretation.

## Supporting information

**S1 Fig. Distribution normality testing for age and Klotho concentrations.** (a) Q-Q plot and (b) density plot for age; (c) Q-Q plot and (d) density plot for serum Klotho.
(TIF)

**S1 Table. Univariate Logistic Regression of the Serum Klotho and Tinnitus prevalence. In (Klotho) represents natural log-transformed serum Klotho concentrations.** The classification of serum Klotho was based on the unweighted lower quartile and median of concentrations (Low: ≤648.75pg/mL; Medium: 648.76~779.8pg/mL; High: >779.8pg/mL). All estimates were weighted to be nationally representative. Abbreviations: OR: odds ratio; CI: confidence interval; BMI: body mass index; PIR: poverty income ratio; PHQ-9: Patient Health Questionnaire-9; PTA: Pure-tone average.
(DOC)

**S2 Table. Associations between serum Klotho and tinnitus severity.** Tinnitus severity was categorized into none (no Problem), mild (small problem) and moderate–severe (moderate/big/very big problem). The classification of serum Klotho was based on the unweighted lower quartile and median of concentrations (Low: ≤648.75pg/mL; Medium: 648.76~779.8pg/mL; High: >779.8pg/mL). All estimates were weighted to be nationally representative.
(DOC)

## Acknowledgments

We express our gratitude for the valuable contributions of the staff and participants involved in the NHANES study.

## Author contributions

**Conceptualization:** Ke-Jiang Du, Shihua Yin.

**Data curation:** Ke-Jiang Du, Bin-Yu Mo.

**Formal analysis:** Tao Hou.

**Investigation:** Long Chen, Deng-Rong Su.

**Methodology:** Ke-Jiang Du.

**Software:** Ke-Jiang Du, Long Chen.

**Supervision:** Bin-Yu Mo, Shihua Yin.

**Validation:** Tao Hou, Long Chen.

**Writing – original draft:** Ke-Jiang Du.

**Writing – review & editing:** Ke-Jiang Du, Shihua Yin.

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
