## [Decision Letter · Decision Letter 0]

Dear Dr. Yin,

Thank you for submitting your manuscript to PLOS ONE. After careful consideration, we feel that it has merit but does not fully meet PLOS ONE’s publication criteria as it currently stands. Therefore, we invite you to submit a revised version of the manuscript that addresses the points raised during the review process.

We look forward to receiving your revised manuscript.

Kind regards,

Calogero Caruso, MD

Academic Editor

PLOS ONE

“This work was supported by the National Natural Science Foundation of China (Grant No. 82160213).”

4. We note that there is identifying data in the Supporting Information file <S1 File.csv>. Due to the inclusion of these potentially identifying data, we have removed this file from your file inventory. Prior to sharing human research participant data, authors should consult with an ethics committee to ensure data are shared in accordance with participant consent and all applicable local laws.

-Location data

Please remove or anonymize all personal, ensure that the data shared are in accordance with participant consent, and re-upload a fully anonymized data set. Please note that spreadsheet columns with personal information must be removed and not hidden as all hidden columns will appear in the published file.

Reviewers' comments:

Reviewer's Responses to Questions

**Comments to the Author**

1. Is the manuscript technically sound, and do the data support the conclusions?

Reviewer #1: Partly

2. Has the statistical analysis been performed appropriately and rigorously?

Reviewer #1: No

3. Have the authors made all data underlying the findings in their manuscript fully available?

Reviewer #1: Yes

4. Is the manuscript presented in an intelligible fashion and written in standard English?

Reviewer #1: Yes

Reviewer #1: Paper by Du et al. evaluate the relationship between Klotho serum levels and the presence of Tinnitus.

Reported results were obtained analyzing data stored in the public database of National Health and Nutrition Examination Survey (NHANES) a survey program of CDC's National Center for Health Statistics (NCHS) US Agency. Results suggest that Low Serum Klotho levels might be associated with the duration of tinnitus (p =0.058). In addition a linear inverse association between klotho (ln values) and tinnitus risk has been observed.

Data are generally clearly stated, even if, some methodological approach might be improved.

In table 1 the term “Race” should be avoided and substituted with “Ethnic origin”.

As reported by authors serum Klotho levels and age are not normally distributed. In consequence authors stratified data into three groups based on the quartile distribution of serum Klotho levels.

Considering that it is well known that klotho levels decrease with age, the evaluation of klotho tinnitus relationship should be made distributing subjects in age quartile distribution. As an example the inverse relationship between Serum Klotho and High-Frequency Hearing Loss has been well demonstrate in a NHANES cohort aged 70-79 years than in other studies.

It is possible that analyzing data for age cohorts a better clear cut evidenced results might be obtained. In addition data on klotho levels reported in table 1 might be biased by age considering that tinnitus positive subjects are significantly older than negative ones (table 1).

It seems that models reported in table 3 are obtained including both tinnitus positive and negative subjects. Maybe the analyses should be performed on positive only to clearly establishing the relationship between reduction of klotho levels and tinnitus.

In discussion authors reported that Klotho downregulation decreases transient receptor potential vanilloid 5(TRPV5) and TRPV6 levels, altering inner ear Ca2+ regulation and causing sensory cell transduction defects that manifest as auditory or vestibular disorders This is a relevant mechanisms that might explain the pathogenetic role of klotho in hearing loss and possibly in tinnitus.

However a discussion on Klotho reduction effect on activity and protein abundance of KCNQ1/KCNE1, a K(+) channel required for proper hearing. In particular considering that, KCN coding gene variants have been recently associated with hearing loss and tinnitus.

Moreover, appears of a certain relevance to discuss some studies that have recently suggested that the Fgf23-Klotho endocrine axis may play a novel role in the inner ear regulating directly or indirectly the kinocilium-stereocilia interaction.

Finally considering that results were obtained analyzing data from a mixed population (Non-Hispanic White, Non-Hispanic Black, Mexican American and Other Hispanic) the limitation 4 “The study was conducted in a single geographical location, which may introduce biases and limit the applicability of the results to broader populations” appears not justified. Instead a comment on the opportunity to complete the study including subjects of Asian ethnicity might be appropriate.

**Do you want your identity to be public for this peer review?** For information about this choice, including consent withdrawal, please see our Privacy Policy

Reviewer #1: No

---

## [Author Response · Author response to Decision Letter 1]

9 Jun 2025

We sincerely appreciate the reviewers’ insightful comments and the opportunity to revise our manuscript (Manuscript ID: PONE-D-25-25271). Please find our point-by-point responses to all journal requirements and reviewers’ suggestions in the "Response to Reviewers" document.

---

## [Decision Letter · Decision Letter 1]

Association of serum Klotho with tinnitus prevalence, duration and severity: A cross-sectional study in middle-aged and older adults

PONE-D-25-25271R1

Dear Colleague,

We’re pleased to inform you that your manuscript has been judged scientifically suitable for publication and will be formally accepted for publication once it meets all outstanding technical requirements.

Kind regards,

Calogero Caruso, MD

Academic Editor

PLOS ONE

Additional Editor Comments (optional):

Reviewers' comments:

Reviewer's Responses to Questions

**Comments to the Author**

Reviewer #1: All comments have been addressed

2. Is the manuscript technically sound, and do the data support the conclusions?

Reviewer #1: Yes

3. Has the statistical analysis been performed appropriately and rigorously?

Reviewer #1: Yes

4. Have the authors made all data underlying the findings in their manuscript fully available?

Reviewer #1: Yes

5. Is the manuscript presented in an intelligible fashion and written in standard English?

Reviewer #1: Yes

Reviewer #1: Paper by Du et al. have been deeply re-managed according to the suggestions. Answers to my questions are fully satisfactory. So, in my opinion, paper is now suitable for publication on PLOS One.

**Do you want your identity to be public for this peer review?** For information about this choice, including consent withdrawal, please see our Privacy Policy

Reviewer #1: No

---

## [Editor Report · Acceptance letter]

PONE-D-25-25271R1

PLOS ONE

Dear Dr. Yin,

I'm pleased to inform you that your manuscript has been deemed suitable for publication in PLOS ONE. Congratulations! Your manuscript is now being handed over to our production team.

Kind regards,

on behalf of

Prof. Calogero Caruso

Academic Editor

PLOS ONE